# A Novel Approach for Micro-Expression Recognition Incorporating Vertical Attention and Position Localization

## Abstract

Micro-expression (ME) is a kind of facial expression that is short-lived and difficult for ordinary people to detect. Micro-expression can reflect the real emotion that people try to hide. It is difficult to identify micro-expression due to the fact that the duration is short and it only involves partial muscle motions, which brings great challenges to the accurate identification of micro-expression. To address these issues, we propose a novel neural network for micro-expression recognition (MER), focusing on subtle changes in facial movements using a CVA (Continuously Vertical Attention) block, which models the local muscle changes with minimal identity information. Additionally, we propose a facial position localization module called FPF (Facial Position Focalizer) based on Swin Transformer, which incorporates spatial information into the facial muscle movement pattern features used for MER. We also proved that including AU (Action Units) can further enhance accuracy, and therefore we have incorporated AU information to assist in micro-expression recognition. The experimental results indicate that the model achieved an average recognition accuracy of 94.35% and 86.76% on the popular CASME II and SAMM micro-expression datasets, improved by 6% and 1.98% compared to state-of-the-art models, respectively.

## 1 Introduction

Micro-expressions are the imperceptible facial expressions that people show when they are deliberately hiding or suppressing their true emotions after being stimulated by the outside world. Micro-expressions mostly come from the subconscious mind and cannot be concealed or suppressed, reflecting a person's real thoughts and attitudes at a certain moment(Davison et al., 2016). A micro-expression contains three stages: onset, apex and offset, as the muscle intensity begins to increase, reaches the peak, and finally goes back to the neutral state. It is widely used in fields such as psychotherapy (Zhu et al., 2017), criminal investigation and national defence and security(See et al., 2019). Therefore, the use of computer technology to assist in capturing facial information has become an inevitable trend. Automatic recognition of micro-expressions is not only a challenge in computer technology, but also involves a number of fields such as physiology and psychology.

According to the different ways of feature extraction, the current micro-expression recognition methods can be mainly divided into two categories: traditional manual features and deep learning. The early recognition methods are micro-expression recognition methods based on manual features, mainly local binary patterns (LBP)(Wang et al., 2015) and optical flow(Liu et al., 2015) methods. These manual feature extraction methods require large inputs from experts might be subjective at some point, and are not able to extract the deeper spatial and temporal information of the image sequence, resulting in the unsatisfying accuracy of MER. In recent years, as deep learning has shown powerful advantages in image processing, more and more researchers have made great progress in micro-expression recognition based on neural networks (Takalkar et al., 2018). Gan et al. (2019) introduced a feature extractor, which combined the features of optical flow and convolutional neural network .Zhao et al. (2021)proposed a 3D CNN-based learning method for spontaneous micro-expression recognition, which outperformed traditional methods and other deep learning methods, and provided new insights on how to use scarce data for MER recognition.

However, the above algorithms do not pay attention to the identification problem, that is the model might learn identity information instead of the feature itself during training. To address this issue,

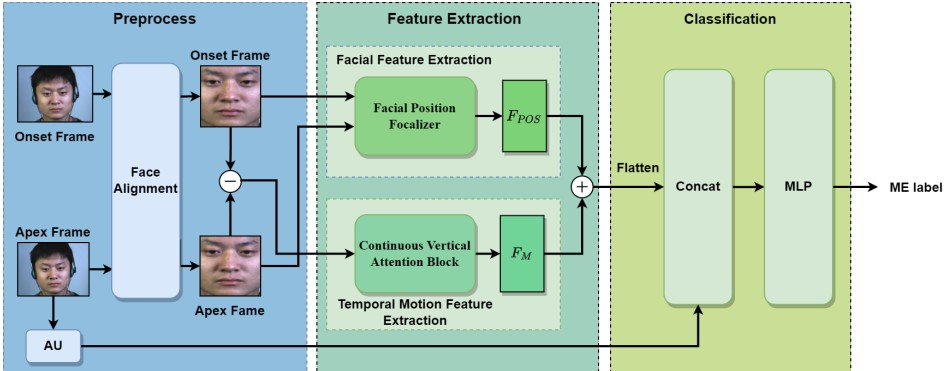

Figure 1: Diagram of our proposed model. It consists of three stages and two main feature extraction part. MLP stands for multi-layer perceptron and ME label refers to micro-expression label

Li et al. (2022) constructed MMNet to extract the motion and location information of the muscles in the relevant region using the difference between the onset and apex frame, which reduces the influence of identity information on recognition and further improves the accuracy of micro-expression recognition. While MMNet innovatively employs motion-based input and incorporates a continuous attention mechanism to capture facial muscle movement features, it is constrained by the fundamental characteristics of ViT, which only captures local relationships, thereby failing to model the essential distant dependencies required for MER. Additionally, the use of single-frame images as position information embeddings inevitably results in information loss.

To this end, inspired by MMNet, we propose a MER model based on continuous spatio-temporal attention blocks, focusing on subtle facial changes. Our framework consists of three parts: preprocessing, feature extraction, and classification. The model consists of two streams, the main stream and the auxiliary stream. In the stage of motion feature extraction, before the results are input into the continuous vertical attention (CVA) block, we subtract the starting frame from the apex frame to perform the subtraction operation. This subtraction operation focuses on modeling the difference between the two frames, which mainly reflects the muscle movement of the face. The main stream CVA takes the motion difference information obtained by the above method as input and leverages the continuous spatio-temporal attention model to extract the spatio-temporal information of muscle motion. The sub-stream takes the starting frame as input information, and through the FPF module, it locates the facial position information in space, pays attention to the local facial feature changes, and finally inputs the fusion information into the classifier to classify the micro-expressions.

To sum up, the main contributions of this paper are:

- We identified the significant contribution of facial muscles in the vertical direction to micro-expression recognition. Leveraging this insight, we designed the Continuous Vertical Attention module (CVA) to enhance long-range dependency modeling capabilities for the extraction of facial muscle movement features.

- We introduced the Facial Position Focalizer (FPF) module to incorporate facial position information using Swin Transformer, effectively suppressing individual variations in background activations. Inspired by the nature of micro-expression muscle movements, we added AU embeddings to assist the network in focusing on active facial regions.

- We integrated the proposed modules into a dual-stream network and evaluated it on the popular CASME II and SAMM datasets. Our method exhibited significantly higher accuracy and F1-scores compared to state-of-the-art methods.

## 2 RELATED WORK

Most of the existing micro-expression recognition methods directly extract features from image sequences that only contains appearance information, including optical flow-based methods, texture description-based methods and deep learning-based methods, but these methods ignore the muscle motion encoded in Action Unit (AU), resulting in the redundant information like identity in the

features. Xie et al. (Xie et al., 2020) modeled different AUs based on relational information, and integrated the AUs recognition task with MER. Although their primary network employs a Graph Attention Convolutional Neural Network, which is not ideally suited for small dataset MER tasks and results in lower recognition accuracy, the integration of AUs information still successful improved their model's performance. Inspired by their work, we also incorporate AUs in our model input, providing necessary information about muscle movement. Liong et al. (2018) propose a model with the onset and apex frames as the input instead of the image sequence. By only using two frames to represent the expression, it avoids introducing complex time series algorithms while keeping the accuracy compared to using image sequences. Based on the aforementioned reasons, our model has three inputs: onset frame, apex frame, and AUs.

In MMNet Li et al. (2022), adaptive attention mechanism inspired by CBAM (Convolutional Block Attention Module) is adopted to extract muscle movement information. However, due to the global focus of CBAM, the model may inadvertently notice identity-related information unrelated to MER. Considering that vertical facial muscle movement plays a more important role in micro-expression recognition (MER), we developed a continuous vertical attention block, which focuses on capturing vertical movement information, and generates an attention map for each layer by merging information from the previous layer. According to the follow-up experimental results, it can be found that applying the attention mechanism only in the vertical direction can help the model focus on muscle movement, thus enhancing the performance of the model.

Facial positioning is a technology used to identify and locate facial features. By determining key positions such as eyes, nose and corners of the mouth, we can analyze and capture the information of muscle changes more accurately. By using the embedded information of these facial areas, we can better understand the dynamic characteristics of the face, thus improving our perception of facial expressions and actions. In previous work, some methods used Graph Convolution Network (GCN) (Kipf & Welling, 2016) as the resolution to represent key points, but GCN assumed that each node's representation was only related to its neighbors, ignoring distant nodes or global information, which limited GCN's ability to deal with the structural characteristics of global graphs. ViT(Dosovitskiy et al., 2020) is used for position embedding in MMNet, where the self-attention mechanism is usually used to capture the local and global relations in the input sequence or image. However, it may have difficulties in capturing long-distance dependencies and global consistency. Considering that positional information from distinct facial components holds varying degrees of significance, and micro-expressions are typically constituted by subtle movements distributed across diverse facial regions, the process of assigning weight to features from various locations places a substantial demand on the model's capacity to efficiently model long-range dependencies. In this respect, Swin Transformer Liu et al. (2021) is more suitable for this task than ViT. The shift window mechanism in Swin Transformer combines spatial position shuffling when calculating self-attention, so it can effectively capture long-distance dependencies. Therefore, we use Swin Transformer to extract facial features and focus on some important areas, and we also apply feature maps of different scales at different depths of the network, which can better adapt to micro-expressions of different durations and patterns.

## 3 METHOD

Our architecture is shown in Figure 1. It takes the onset frame, apex frame, and corresponding Action Unit (AU) information of a micro-expression sequence as input and outputs predicted micro-expression labels. The structure consists of three main stages: Preprocessing, Feature Extraction, and Classification. In the Feature Extraction section, it has two streams: Facial Feature Extraction and Temporal Motion Feature Extraction. In Facial Feature Extraction, we extract information from the onset frame and the apex frame separately by a Swin Transformer encoder and then add them together. Temporal Motion Feature Extraction focuses on extracting information regarding the subject's facial muscle motion. We subtract the onset frame from the apex frame before inputting the result into the continuous vertical attention (CVA) block. This subtraction operation focuses on modeling the difference between the two frames, which primarily reflects the muscle motion on the face. Finally, we fuse facial and motion features by performing a summation operation and then combine the AU information by concatenation. A multi-layer perceptron (MLP) receives all the information and outputs the most likely emotion label. We will provide a detailed explanation of these steps in the following sections.

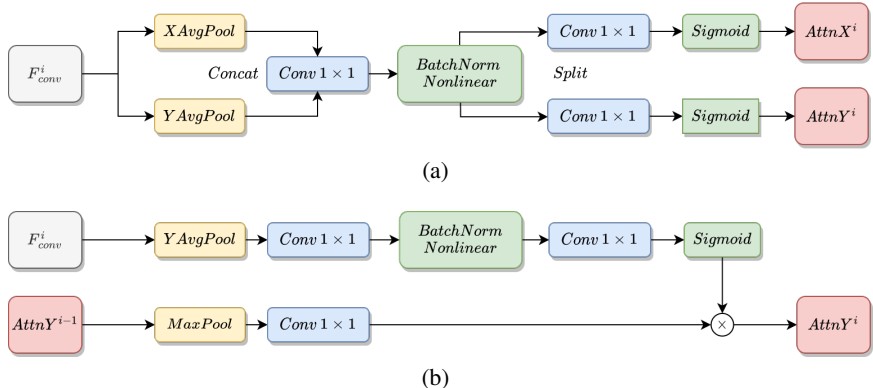

Figure 2: the spatial attention module of CA module(a) is compared with our CVA module(b)

## 3.1 CONTINUOUS VERTICAL ATTENTION BLOCK

In MMNet, an adapted attention mechanism that is inspired by CBAM was employed to extract muscle motion information. However, due to CBAM's characteristic of computing attention globally, the model may inadvertently pay attention to identity-related information that is irrelevant to MER. Given the fact that vertical facial muscle movement plays a more significant role in MER than horizontal movement, we develop a Continuous Vertical Attention Block which concentrates on capturing vertical movement information and generates the attention map for each layer by incorporating information from the previous layer. Inspired by the coordinate attention module (CA) depicted in Figure 2(a), we utilize a separate attention mechanism along the y-axis to compute the necessary vertical weights. As shown in Figure 2(b), we introduce the attention maps of previous layers to reweight the current attention and use max pooling to obtain the most prominent feature. The CVA module is defined as follows:

$$
\begin{aligned}
AttnY^i &= F_M{}^i(F_{conv}^i, AttnY^{i-1}) \\
&= \sigma(f_2^i(F_{act}(BN(f_1^i(P_{AY}(F_{conv}^i)))))) \bigotimes f_{1\times1}^i(P_M(Attn^{i-1})),
\end{aligned}
\tag{1}
$$

with

$$
F_{conv}^i = f_{1\times1}^i\left(f_{3\times3}^i\left(F^i\right)\right) + f_{1\times1}^i\left(F^i\right),
\tag{2}
$$

$$
F_{act}(x) = x \cdot ReLU(x+3)/6,
\tag{3}
$$

where $F_M^i$ is the CVA module of the $i^{th}$ CVA block and $AttnY^i$ represents the attention map computed by the $i^{th}$ layer of the CVA module. $F^i conv$ serves as the input to the CVA module, representing feature maps extracted through two layers of convolutional networks and a residual structure. Letter $\sigma$ is the sigmoid function. $P_{AY}$ and $P_M$ denote adaptive average pooling operations along the height dimension and max pooling operations along both the height and width dimensions, respectively. Both $f_1^i$ and $f_2^i$ are convolution operations with a kernel size of 1 at the i-th layer. $f_1^i$ is employed for dimensionality reduction of the feature maps, while $f_2^i$ is utilized for dimensionality restoration. Specifically, the dimensionality reduction factor is determined as max$\{8, C/32\}$, where $C$ represents the channel dimension of the feature maps. $f_{1\times1}^i$ and $f_{3\times3}^i$ represent a convolution operation at the i-th layer with the kernel size of 1 and 3, respectively. BN stands for the batch normalization and ACT means the non-linear activation function. $\bigotimes$ is the product on the element-wise which can reweight the attention map of current layer by introducing the attention map of last layer while $F^i$ represents the input of the i-th CVA block.

In the CAV block, we consecutively stack four CAV modules as illustrated in Figure 3. Due to the continuous attention mechanism embedded within these modules, the network gradually shifts its focus towards the facial muscle motion that are more beneficial for MER. Ultimately, the CAV block

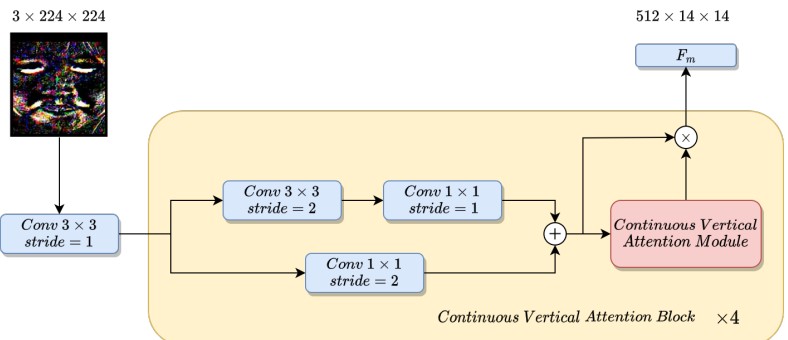

Figure 3: Schematic diagram of continuous vertical attention block

yields a feature map $F_M$ with dimensions of $512 * 14 * 14$. The CAV block is formally formulated as,

$$F_B(F^i, AttnY^{i-1}) = F^i_{conv} \odot F_M{}^i(F^i_{conv}, AttnY^{i-1}), \tag{4}$$

where $F_B$ means the continuous vertical attention block and $\odot$ represents the broadcast element-wise multiplication, which makes every column of the motion-pattern feature maps reweighted by the vertical attention maps.

## 3.2 FACIAL POSITION FOCALIZER

Inspired by the concept of positional information embedding in MMNet, we introduced the FPF (Facial Position Focalizer) module based on Swin Transformer. Considering that within MER, positional information from distinct facial components holds varying degrees of significance, and micro-expressions are typically constituted by subtle movements distributed across diverse facial regions, the process of assigning weight to features from various locations places a substantial demand on the model's capacity to efficiently model long-range dependencies. The shifted window mechanism in Swin Transformer incorporates spatial position shuffling when computing self-attention, enabling the effective capture of relationships between long-distance dependencies. Unlike the single-stage downsampling used in ViT, the Swin Transformer achieves downsampling through a patch merging layer in every stage. This approach effectively handles higher-resolution images while preserving more information. Therefore we introduce the FPF module based on Swin Transformer. As shown in Figure 1, we utilize the difference between the apex frame and onset frame to learn facial motion features. Considering that the apex frame and onset frame may not be strictly aligned, we input each frame separately into the FPF module to learn facial position information. Since our goal is to focus on learning facial position information rather than fine-grained features, we employed a Swin Transformer model with three layers and a depth configuration of [2, 2, 6]. Subsequently, we reshape the onset frame and apex frame into a series of 128 flattened 2D patches. After passing through three downsampling layers, their channel dimensions match those of the $F_M$. After that, the Swin Transformer encoder will receive these patches and learn the relationship between them. Then, the Swin Transformer will yield two feature maps with dimensions of $512 * 196$, which are then reshaped to $196 * 14 * 14$ to match the dimensions of the $F_M$. Finally, the two feature maps generated separately from the onset frame and apex frame are integrated by element-wise addition to consolidate.

## 3.3 AU EMBEDDING

In MER tasks, AUs provide more useful information about facial movements compared to image sequences. They serve as valuable cues and supplements for MER. Additionally, AUs also represent the activity range of muscles, forcing the model to better focus on these specific facial regions. Hence, we introduce a binary 2D vector of length 21, representing the activity states of twenty-one AUs (0 indicating inactive, and 1 indicating active) as an activation area embedding which guides the model's attention to a more critical facial region. As depicted in Figure 1, after adding $F_{POS}$ and $F_M$ together, the resulting feature map is flattened and then concatenated with the AU information. Finally, these features are processed through an MLP network to output the probability of emotion labels.

## 4 EXPERIMENTAL RESULTS

In this section, we will illustrate the promising effect of our model through experiments. First,we introduce the datasets and setup. Second, we will provide the details of the experiment implementation. Then, the influence of each part of the model will be explored with extensive ablation study. Further, we compare our result with state-of-the-art in MER.

### 4.1 DATASETS

We validate our method on two popular classic datasets to prove its impact.

**CASME II(Chinese Academy of Sciences Micro-Expression II)**(Yan et al., 2014)is a dataset of spontaneous micro-expressions collected from 26 participants. It contains 255 videos, recorded using a high-speed camera with a frame rate of 200 fps. The videos are labeled with 5 emotion categories: happiness, disgust, repression, surprise, and others.

**SAMM (Spontaneous Micro-Facial Movement)**(Davison et al., 2016) is another dataset of spontaneous micro-expressions collected from 32 participants,containing 159 micro-expression videos at 200 fps. Consistent with most other MER methods, we conducted experiments using the typical five emotion labels (happiness, anger, contempt, surprise, and others).

It's worth noting that in both the CASME II dataset and the SAMM dataset, more detailed annotations for Action Units (AUs) have been provided[1]. However, in our experiments, we utilized only the binary 01 information representing whether AUs were active or not[2]. Similar to most other approaches, we employed the widely accepted leave-one-subject-out (LOSO) cross-validation method. This method involves using each subject in turn as the test set while utilizing the remaining subjects as the training data which effectively mitigates subject bias and allows for the evaluation of the generalization performance of various algorithms.

### 4.2 EVALUATION METRIC

In our experiments, we utilize accuracy and F1-score as the evaluation metrics to assess the performance of our model. Although accuracy is the primary metric for assessing performance in most classification tasks, it can be influenced by data imbalance when present. Indeed, the F1-score, by taking into account the total True Positives (TP), False Positives (FP), and False Negatives (FN), is less influenced by imbalanced classes and provides insight into the true performance of a classification system. We calculate the average F1-score across the 5 emotional labels:

$$F1 - score = 2 \times \frac{P \times R}{P + R} \tag{5}$$

with

$$P = \frac{TP}{TP + FP}, R = \frac{TP}{TP + FN}. \tag{6}$$

### 4.3 IMPLEMENTATION DETAILS

We use Dlib version 19.7.0 and OpenCV version 3.4.9.33 for face alignment and image cropping. The images are resized to $224 * 224$ followed by three data argumentation techniques: horizontal flipping (probability 0.5), random cropping (padding 15), and random rotation($0°$ to $3°$).

The number of Swin Transformer encoder blocks is set to 3, with the depths and number of heads per stage being 2, 2, 6 and 4, 8, 16 respectively. We employe gradient accumulation method to reduce the GPU memory footprint of the model. Each mini-batch had a size of 16, and the gradient update was performed every two mini-batches.

For the training parameters, the learning rate was initialized to 0.0008 and exponentially decayed during the first 50 epochs out of 75 epochs in total. We use AdamW to optimize the network with a weight decay of 0.6. All the experiments were carried out on RTX 3080 graphics card with version 2.0.0 of Pytorch toolbox.

---

[1]e.g. R20+ representing the right corner of the mouth being pulled upward

[2]e.g. 01001 indicating AU2 and AU5 being active

| Method | Accuracy(%) | F1-score |
|--------|-------------|----------|
| DSSN (2019) | 71.19 | 0.7297 |
| TSCNN (2019) | 80.97 | 0.8070 |
| Dynamic (2020) | 72.61 | 0.6700 |
| Graph-TCN (2020) | 73.98 | 0.7246 |
| SMA-STN(2020) | 82.59 | 0.7946 |
| AU-GCN (2020) | 74.27 | 0.7047 |
| GEME (2021) | 75.20 | 0.7354 |
| MERSiamC3D (2021) | 81.89 | 0.8300 |
| MMNet (2022) | 88.35 | 0.8676 |
| AMAN (2022) | 75.40 | 0.7100 |
| MiMaNet (2021) | 79.90 | 0.7590 |
| $\mu$-BERT (2023) | 83.48 | 0.8553 |
| Ours | **94.35** | **0.9402** |

Table 1: Comparison between several SOTA methods and our model on CASME II(5 classes)

| Method | CASME II(5 classes) | | SAMM | |
|--------|-------------|----------|-------------|----------|
| | Accuracy(%) | F1-score | Accuracy(%) | F1-score |
| Baseline(ResNet) | 81.12 | 0.7582 | 73.53 | 0.6345 |
| CVA block | 84.68 | 0.8284 | 76.69 | 0.6633 |
| FPF block | 89.52 | 0.8543 | 81.20 | 0.7067 |
| AU Embedding | 89.92 | 0.8639 | 81.95 | 0.7428 |
| CVA block+FPF block + AU Embeddir | 94.35 | 0.9402 | 86.73 | 0.8171 |

Table 2: Evaluate the contribution of AU embedding, CVA block and FPF block to the enhancement of network performance of basic network ResNet-18(He et al., 2016)

## 4.4 RESULTS

The comparison between several SOTA methods and our model demonstrate the superiority of our method. The experiments on every evaluation standard indicate our method achieve great advancement in micro-expression recognition. From Table 1, we can see that the model we devised has outperformed all previous methods in terms of both accuracy and F1-score. After introducing the CA and FPF modules, which possess stronger long-range dependency modeling capabilities compared to the PC module and CA module, our model's performance significantly surpasses that of MMNet. To specify further, in the five-expression category task, our accuracy and F1-score outperformed MMNet by 6% and 7.26% in the CASME II dataset. In the latest $\mu$-BERT, a Diagonal Micro-Attention (DMA) is employed to detect tiny differences between two frames with the introduction of a new Patch of Interest (PoI) module to localize and high-light micro-expression interest regions.However, thanks to our vertical attention mechanism, our model excels in focusing on facial expression information that contributes more significantly to Micro-Expression Recognition (MER) compared to $\mu$-BERT. This enables us to achieve higher accuracy rates than $\mu$-BERT on SAMM and CASME II with 1.98% and 10.87% respectively.

## 4.5 ABLATION STUDY

To validate the effectiveness of the individual components of our model, we conducted a series of ablation experiments.

### 4.5.1 EFFECTIVENESS OF DIFFERENT BLOCKS IN THE MODEL

To ensure a more objective comparison, we employed the ResNet-18 network as a baseline and individually incorporated AU embedding, CVA block, and FPF block to assess their respective contributions to enhancing the network's performance. As depicted in Table 2, it is evident that all three components significantly enhance the performance. When we combine all three modules together, in the five-expression category task, our accuracy and F1-score surpass the baseline by 13.23% and 18.2% in the CASME2 dataset and by 13.2% and 18.26% in the SAMM dataset, respectively.

| Method | CASME II(5 classes) | | SAMM | |
|---|---|---|---|---|
| | Accuracy(%) | F1-score | Accuracy(%) | F1-score |
| Vertical attention(Y) | 94.35 | 0.9402 | 86.73 | 0.8171 |
| Horizontal attention(X) | 84.68 | 0.8234 | 78.20 | 0.6927 |
| Both direction attention(X+Y) | 92.74 | 0.9276 | 85.71 | 0.7845 |

Table 3: Performance comparison of vertical attention with horizontal attention and Both direction attention

| Method | CASME II(5 classes) | | SAMM | |
|---|---|---|---|---|
| | Accuracy(%) | F1-score | Accuracy(%) | F1-score |
| Independent attention | 91.94 | 0.921 | 82.71 | 0.7892 |
| Continuous attention | 94.35 | 0.9402 | 86.73 | 0.8171 |

Table 4: Comparison between independent attention and continuous attention

### 4.5.2 EFFECTIVENESS OF OUR CONTINUOUSLY VERTICAL ATTENTION MODULE

To illustrate the superiority of our proposed CVA module, we conducted experiments using only the vertical attention mechanism, the horizontal attention mechanism, and both directional attention mechanisms separately. It's worth noting that in all three of these configurations, we included the FPF block and AU embedding to ensure the objectivity of the comparison. As seen in Table 3, the use of only the vertical attention mechanism results in the model generating more precise attention maps, demonstrating superior performance compared to the other two configurations.

Additionally, through experimentation, we have demonstrated that introducing the attention map from the previous layer as prior knowledge can effectively enhance the model's performance, as illustrated in Table 4.

### 4.5.3 COMPARISON BETWEEN SINGLE & DUAL SUBBRANCH

To verify that Swin Transformer is better at capturing long-term dependencies in MER tasks compared to ViT, we conducted a series of experiments in which AU embedding, and the CVA block were incorporated. As shown in Table5, the FPF module based on Swin Transformer outperformed ViT significantly on both datasets.

In the third section, we propose that by inputting image information from both the onset and apex frame, we can reduce the impact of imperfect facial alignment and enable the model to extract more comprehensive facial position information. As depicted in Table6, through a comparison between single-frame input and dual-frame input, our hypothesis is empirically validated.

### 4.5.4 IMPROVEMENTS FROM AU EMBEDDING

We compared the impact of adding AU embedding to the model, as shown in Table7. The significant effect on classification performance confirms our belief that AU information can assist the model in focusing on crucial facial regions required for MER.

| Method | CASME II(5 classes) | | SAMM | |
|---|---|---|---|---|
| | Accuracy(%) | F1-score | Accuracy(%) | F1-score |
| FPF block based on ViT | 91.12 | 0.9014 | 83.46 | 0.8025 |
| FPF block based on Swin Transfomer | 94.35 | 0.9402 | 86.73 | 0.8171 |

Table 5: Performance comparison between FPF block based on ViT and FPF block based on Swin Transformer

| Method | CASME II(5 classes) | | SAMM | |
|---|---|---|---|---|
| | Accuracy(%) | F1-score | Accuracy(%) | F1-score |
| Apex input | 90.73 | 0.9113 | 84.96 | 0.7845 |
| Onset input | 91.13 | 0.9112 | 85.71 | 0.7932 |
| Dual-frame input | 94.35 | 0.9402 | 86.73 | 0.8171 |

Table 6: Performance comparison between single-frame input and double-frame input

| Method | CASME II(5 classes) | | SAMM | |
|---|---|---|---|---|
| | Accuracy(%) | F1-score | Accuracy(%) | F1-score |
| Wihout AU embedding | 89.52 | 0.9014 | 81.20 | 0.7519 |
| With AU embedding | 94.35 | 0.9112 | 86.73 | 0.8171 |

Table 7: Performance comparison of AU embedding

## 5  CONCLUSION

In this paper, we design a new dual-stream MER network, the main branch based on the proposed CVA module focuses on learning the motion pattern characteristics from the onset frame to the apex frame, while the sub-branch based on the FPF module focuses on generating facial position embedding for position calibration, and we add AU to improve the accuracy. Experiments show that the performance of our model on CASME II and SAMM data sets is far superior to the existing technology.

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
