# OpenReview forum: "A Novel Approach for Micro-Expression Recognition Incorporating Vertical Attention and Position Localization"
_ICLR.cc/2024/Conference — Submitted to ICLR 2024_

### Official Review · Reviewer_Vkxu · 2023-10-29

**Soundness:** 2 fair
**Presentation:** 3 good
**Contribution:** 2 fair
**Rating:** 3
**Confidence:** 4

**Summary:**

This paper deals with the micro-expression recognition problem. It proposes a Continuously Vertical Attention (CVA) block to model the local muscle changes excluding the identity information. It proposes a Facial Position Focalize (FPF) module to incorporate spatial information and also utilizes AU information to further improve performance. Experiments on two micro-expression datasets validate that the proposed method achieves superior performance than other models.

**Strengths:**

1. The motivation of the proposed method is reasonable. It proposes a CVA block to model the continuous local muscle changes.

2. The paper is easy to follow and the proposed method is easy to understand.

3. Experiments on two datasets show that the proposed method achieves superior accuracy and F1-score compared with other methods.

**Weaknesses:**

1. The novelty is limited. As far as I'm concerned, the continuous attention block and position calibration module are both proposed in the MMNet paper. It also utilizes the subtraction between the apex and onset to determine the local movement of muscles. Thus, the difference between this work and the previous work lies in that this paper uses vertical attention instead of both directions, which is not quite novel.

2. This paper does not contain any visualization results. For example, what is the difference between vertical attention and both directions regards the attention maps? What are the feature distributions before and after using the AU information?

3. Experiments on other datasets and other metrics are required. For example, MMNet also carried out experiments on the MMEW dataset. Furthermore, in paper [1], it carried out experiments on CASME3 and SMIC datasets. The performance of the two datasets is not enough to validate the effectiveness of the proposed method in the micro-expression recognition field. The authors should consider using the metrics of UF1 and UAR to make fair comparisons with the paper [1].

[1] Micron-BERT: BERT-based Facial Micro-Expression Recognition

**Questions:**

1. Why using vertical attention instead of both-direction attention could improve the performance? Is there any justice or similar phenomenon in other papers? To me, using vertical attention means some information in the horizontal direction is lost.

2. In Table 2, It seems to me the most improvement of the performance is brought by the AU Embedding. What is the performance using CVA block + FPF block without AU Embedding? The last column has a typo, embeddir should be embedding.

---

### Official Review · Reviewer_Bj7L · 2023-11-01

**Soundness:** 3 good
**Presentation:** 3 good
**Contribution:** 2 fair
**Rating:** 5
**Confidence:** 5

**Summary:**

The authors propose a micro facial expression recognition framework that includes preprocessing, feature extraction, and classification. They propose a continuous vertical attention module that aims at local muscle movements and a facial position localizer that focuses on spatial information. They also included AUs for better information. Experiments are performed on CASME II and SAMM datasets and compared with previous methods.

**Strengths:**

The paper targets an important issue of micro facial expression recognition. It is well-written step by step and easy to follow. The encoding of vertical facial muscle movement is interesting. Multiple ablations were conducted to show the importance of each component. The proposed method gets better accuracy on two datasets.

**Weaknesses:**

The paper uses extra information( apex, onset frames and AUs) and in experiments, it is not clearly mentioned which previous work uses that information. My detailed comments and questions are as follow.
1> Apex frame and onset frame are used to extract information about microexpression. Extracting these frames from a sequence of frames is itself a task. How do you compare your work with other methods that use whole sequences?
2> AUs reveal a lot of information about expression. Getting AU information from images is again a challenging task. Authors are using directly available AUs in their method. How do you compare your method with other works that are using this readily available information?
3> This work subtracts two frames for difference information. Did the authors encounter a situation where two frames are not aligned perfectly? Will it affect the further process and How do you encounter that?
4> Since this work targets the removal of identity information, it will be better to mention papers that also target identity removal. (if possible experimental comparison too)
5>How do the authors justify that vertical facial muscle movements are more important than horizontal ones (other than experiments)?
6? In section 3.3 authors mention ‘lenght’ 21 2D vector. What do you mean by length in a 2D vector?
7> Is it possible to give some visuals of the vertical attention module?
8>There are multiple space and dot errors in the paper. Please proofread carefully.

**Questions:**

Please see weakness section.

---

### Official Review · Reviewer_gEuL · 2023-11-06

**Soundness:** 3 good
**Presentation:** 3 good
**Contribution:** 2 fair
**Rating:** 6
**Confidence:** 3

**Summary:**

The authors propose a pipeline for micro-expression recognition (MER), which are brief facial expressions often challenging to identify due to their short duration and partial muscle movements. Their approach includes a Continuously Vertical Attention (CVA) block to capture subtle facial muscle changes without focusing on identity information and a Facial Position Focalizer (FPF) module based on the Swin Transformer. The authors also integrated  Action Units (AU) information into the pipeline to enhance accuracy. Their experiments show that the proposed model achieved significant improvements compared to the state-of-the-art techniques for microexpression recognition. Further ablation studies signified using CVA, FPF and action unit information.

**Strengths:**

The authors attributed the novelty of the work to the use of a vertical attention module, a facial position focalizer using a Swin transformer and the use of AU embeddings to focus on active facial regions due to muscle activations. The modules were integrated into a dual-stream network to recognize microexpressions. The proposed ensemble was trained and validated on CASMEII  and SAMM datasets.

The proposed pipeline is novel, introducing a vertical attention module. However, the use of difference images and Swin transformer to extract feature representations and AU integration for embeddings are existing mechanisms in literature. Their integration with VAC does bring in an aspect of extended feature representation that is beneficial for better recognition accuracies.

**Weaknesses:**

The authors highlight the significant contribution of facial muscles in the vertical direction and other submodules for identifying microexpressions. However, the authors needed to substantiate this understanding within the experimental setup. The increase in accuracy and F1 score, is it statistically significant? Were these results cross-validated, and if so, what is the expected deviation from the reported results?

It also needs to be clarified whether the authors handled bias in the labels for the reported results. This information would be crucial to validate the results reported.

It would be beneficial to generate heat maps for the test samples to understand the areas of significance utilized by the pipeline for prediction purposes. This information may be included in the Appendix if necessary.

**Questions:**

1. Will the reported results change for vertical attention vs horizontal attention vs combined if the bias (if any) was handled? What was the size and distribution of the test set?

2. Is the difference in reported results significant?

3. If yes, what is the expected deviation from the mean during multiple folds of the training and testing?

---

### Official Review · Reviewer_d3J6 · 2023-11-28

**Soundness:** 2 fair
**Presentation:** 3 good
**Contribution:** 2 fair
**Rating:** 5
**Confidence:** 5

**Summary:**

The presented work proposes a neural network addressing micro-expression recognition task. The method utilizes  a multi branch approach, extracting motion patterns from the difference between the onset frame and the apex frame and generates facial features  incorporating facial position information using Swin Transformer. Features representing facial feature movements are enhanced using Continuous Vertical Attention Module. The method is validated on two public datasets and the ablation study is performed for evaluation of each introduced block.

**Strengths:**

1. The related work section is comprehensive, clearly introducing previous solutions and listing their limitations
2. The performed ablation study covers all introduced blocks showing their relevance to the MER task.
3. The selection of Swin Transformer is justified in the presented experiments, showing its advantage over the ViT and supporting the claim that the shift window mechanism is crucial for capturing long-distance dependencies.

**Weaknesses:**

1. The novelty of the proposed model is limited. The method is very similar to MMNet that also uses two branches, apex and onset frame difference as the input to the motion pattern block, Continuous Attention module and the mechanism for position calibration, which is slightly different but serves similar purpose.
2. Figure 2a CA module looks different than the CA module presented in the original MMNet work. In fact, the CA module in the MMNet work closely resembles the proposed CVA module presented in Fig. 2b. The math for the CA ad CVA modules is also very similar, indicating high similarity between both blocks. The only difference seems to be in the use of the vertical direction only.
3. Results for 3 classes should be also provided to ensure detailed comparison with previous methods.

**Questions:**

1. What would be the gain in using detailed annotation of AU instead of just binary information?
2. Are you planning to make the implementation publicly available?
3. Please provide visualization of attention maps to show that the method pays attention to micro expressions. Have you performed such analysis?
4. Were all methods retrained in the same way, using the LOSO cross-validation method?

---

### Meta-Review · Area_Chair_2RSm · 2023-11-30

**Metareview:**

**Summary:** The authors propose a framework for micro-expression recognition (MER) that addresses the challenges of identifying brief facial expressions.  Their pipeline includes a Continuously Vertical Attention (CVA) block to capture subtle muscle changes, a Facial Position Focalizer (FPF) module for spatial information, and integration of Action Units (AUs) to enhance accuracy.  Experimental results on selected micro-expression datasets show that the proposed model outperforms existing techniques in MER.  However, the paper has several weaknesses, including a lack of detail in the experimental setup to explain the contribution of facial muscles and vertical submodules, make the reviewers to question the results.

While the paper's elements have potential, it lacks a clear presentation and thorough experimental evidence.  The authors did not reply to the reviewers' concerns.  I see a trend recommending the rejection of the paper.  During the post-rebuttal discussion, the more excited reviewer gEuL also recommended the rejection of the paper.

**Strengths:** The paper introduces a novel pipeline for micro facial expression recognition.  It utilizes a vertical attention module, a facial position focalizer based on the Swin transformer, and incorporates Action Unit (AU) embeddings to focus on active facial regions.  The proposed method achieves superior accuracy and F1-score compared to the other selected methods on two datasets.  While the integration of the vertical attention module is novel, the use of difference images, Swin transformer, and AU integration have been explored in existing literature.  However, their integration with the vertical attention module brings extended feature representation, leading to improved recognition accuracies.

**Weaknesses:**  The main weaknesses identified by the reviewers include the lack of detail in the experimental setup to explain the contribution of facial muscles and vertical submodules.  The handling of bias in the labels is also unclear, and the suggestion to use heat maps during prediction is made.  Questions are raised about the robustness of the reported results under different treatments of biases, test set size, and distribution.  The missing comparisons and key questions make it difficult to evaluate the work, and the novelty of the paper compared to previous research is questioned.  Visualization results are also missing, and the paper requires attention to copy editing issues. Additionally, a comparison with other papers on the method of identity removal would provide further insight. Overall, the paper lacks clear presentation and convincing evidence.

**About late review:**  There was a late review added by Reviewer d3J6 to the paper during the post-rebuttal discussion.  This review was ignored during the discussion as well as during the assessment of the paper described above.

**Justification For Why Not Higher Score:**

The reviewers raised critical concerns regarding the novelty and experimental aspects of the paper, particularly the lack of comparison against other strong baselines. Furthermore, the reviewers found the results to be unconvincing, leading them to not recommend the paper for publication in its current form.

**Justification For Why Not Lower Score:**

N/A

---

### Decision · Program_Chairs · 2024-01-16

Reject